# In Silico Maturation of a Nanomolar Antibody against the Human CXCR2

**DOI:** 10.3390/biom12091285

**Published:** 2022-09-13

**Authors:** Damiano Buratto, Yue Wan, Xiaojie Shi, Guang Yang, Francesco Zonta

**Affiliations:** Shanghai Institute for Advanced Immunochemical Studies, ShanghaiTech University, Shanghai 201210, China

**Keywords:** in silico affinity maturation, MM-PBSA (Molecular Mechanic/Poisson-Boltzmann Surface Area), CXCR2, monoclonal antibodies

## Abstract

The steady increase in computational power in the last 50 years is opening unprecedented opportunities in biology, as computer simulations of biological systems have become more accessible and can reproduce experimental results more accurately. Here, we wanted to test the ability of computer simulations to replace experiments in the limited but practically useful scope of improving the biochemical characteristics of the abN48 antibody, a nanomolar antagonist of the CXC chemokine receptor 2 (CXCR2) that was initially selected from a combinatorial antibody library. Our results showed a good correlation between the computed binding energies of the antibody to the peptide target and the experimental binding affinities. Moreover, we showed that it is possible to design new antibody sequences in silico with a higher affinity to the desired target using a Monte Carlo Metropolis algorithm. The newly designed sequences had an affinity comparable to the best ones obtained using in vitro affinity maturation and could be obtained within a similar timeframe. The methodology proposed here could represent a valid alternative for improving antibodies in cases in which experiments are too expensive or technically tricky and could open an opportunity for designing antibodies for targets that have been elusive so far.

## 1. Introduction

Antibodies are large, Y-shaped molecules used by the immune system to identify and neutralize foreign objects such as pathogenic bacteria and viruses. They are composed of two proteins of different molecular weights (heavy chain and light chain) linked together by disulfide bonds. They recognize a unique molecule of the pathogen (the antigen) through their variable region, which is composed of six relatively short loops: three belonging to the heavy chain and three to the light chain. The six variable loops are called complementary determining regions (CDRs), and the possible number of theoretical combinations for such loops can be on the order of 10^20^ [1]. In a healthy individual, the antibody diversity available to the circulating repertoire is vast, perhaps in the region of 10^16^–10^18^, and the number of peripheral blood B cells in a healthy adult human is on the order of 5 × 10^9^, so the circulating B cell population samples only a tiny fraction of this diversity [2]. The large diversity in antibody sequences assures that it is theoretically possible to find at least one antibody that recognizes and tightly binds to any given protein, including those produced by the organism itself. Antibodies can regulate proteins on the cell surface [3,4,5,6] or inside the cell [7], and thanks to these properties, they have been engineered to become not only powerful biotechnological tools but also, more importantly, the largest and fastest-growing class of therapeutic proteins [8,9].

The selection of antibodies for a specific target has been facilitated by the development of monoclonal antibodies (mABs) from combinatorial phage-display libraries [10,11,12,13], a technology that earned the 2018 Nobel Prize for Chemistry. Monoclonal antibodies have several benefits, such as fewer off-target adverse effects, fewer drug–drug interactions, higher specificity, and potentially increased efficacy through targeted therapy. The selection of antibodies from a phage-display library can be performed on cells overexpressing the target protein [14] or using the purified protein or part of it, such as extracellular domains or peptides in the hypothetical or desired binding regions [4,5]. The library is screened for phage binding to the antigen through its expressed surface mAb by a technique called (bio-)panning. Multiple rounds of phage binding to the antigen, washing, elution and re-amplification of the phage binders in E. coli improve the chances of pulling out potentially very rare and potent antigen-binding clones [15].

To further improve the biochemical and biophysical properties of the mAB, a new library can be built from the most promising binders by allowing random mutagenesis of the heavy- and light-chain variable regions by using error-prone PCR. This library undergoes the same panning process described above, and antibodies with desirable characteristics can be extracted from it. This technique is called antibody affinity maturation [16].

Phage-display panning and affinity maturation are potent tools for selecting antibodies with the desired biochemical properties. However, limitations still exist that are essentially linked to the ability (or lack of) to express and purify the target of choice. Alternative strategies based on computer simulations for the improvement of antibodies [17,18,19,20,21,22,23,24,25] or small peptides [26,27] have been successfully explored. However, no golden standard has emerged: accurate calculation of binding free energy between proteins requires long molecular dynamics simulation to accurately sample the phase space [28], while simplified scoring functions [29] produce results that often are system-dependent. The issue is further complicated by the need to explore the vast sequence space of antibody CDR regions.

Nevertheless, the possibility of substituting at least part of the experimental procedure with computer calculation is appealing and worth exploring due to the consequential reduction in cost and time to find a good antibody candidate. In this paper, we propose a method to improve the affinity of an antibody to a target in silico. The binding energy of each antibody was evaluated using the Molecular Mechanics/Poisson–Boltzmann Surface Area (MM-PBSA) method [30,31,32], while the sequence space was explored by using a Monte Carlo method in which the new proposed sequences were accepted or rejected according to the Metropolis algorithm [33,34].

As a proof of concept, we tested our method to improve an antibody that binds to the extracellular N-terminus of the interleukin 8 receptor beta (CXCR2) in the G-protein-coupled receptor family. This antibody was obtained by panning a combinatorial antibody library of phages expressing the single-chain variable fragment (scFv) [35] and using part of the extracellular N-terminus of the CXCR2 as a bait peptide [4]. Our results showed that affinity maturation in silico can be achieved and that the newly designed antibodies could improve the original one, even when the starting antibody had a nanomolar affinity.

## 2. Materials and Methods

### 2.1. Molecular Dynamics Simulations and MM-PBSA

All MD simulations were carried out using the Gromacs 2018 package (University of Groningen, Uppsala University, Uppsala, Sweden, version 2018.8) [36] and the Amber14SB force field [37] following simulation protocols similar to those we used in our previous works [38,39]. System preparation and equilibration were the same for each different antibody–antigen pair. Specifically, each model was solvated with TIP3P water containing Cl^−^ and K^+^ ions at a concentration of ∼0.15 M to mimic the physiological ionic strength. After energy minimization, we performed 200 ps of simulated annealing to allow the side chains to equilibrate (this is important when mutations are introduced). We then performed two short simulations lasting 100 ps, first in the NVT and then in the NPT ensembles, with positional restraints (the position restraint constant was K_PR_ = 500 kJ/mol·nm^2^) on the heavy atoms of the protein. During production runs, temperature (T) and pressure (P) were kept constant at 300 K and 1 atm, respectively, using Berendsen thermostat and barostat [40]. Fast smooth particle–mesh Ewald summation [41] was used for long-range electrostatic interactions with a cut-off of 1.0 nm for the direct interactions.

The structure of abN48 (fragment antigen binding or FAB format) in complex with the CXCR2 N-terminal peptide from residue 9 to 19 (pepN9-19) was derived from the X-ray crystal structure PDB ID 6KVF. CDRs sequences were determined according to the Kabat numbering scheme [42].

This complex was simulated for 26 ns in an unconstrained MD simulation to allow for a proper equilibration of the side chains in the interacting region. We observed that the RMSD of the antibody variable region reached equilibrium after 10 ns, and we chose a random frame after this point as the starting point for the following simulations.

All the mutants of abN48, including those generated by the Monte Carlo method described below, were generated from the equilibrated structure of the complex of abN48 and pepN9-19 by using CHIMERA (Resource for Biocomputing, Visualization, and Informatics (RBVI), University of California, San Francisco (UCSF), CA, USA) [43].

To obtain accurate configuration samplings for calculating the binding free energy given a fixed amount of computational power it is preferable to run independent replicas of short simulations rather than a single longer one [30]. This also proved true in our system by comparing the results obtained from a single 25 ns MD trajectory with those obtained from 10 replicas of a 2 ns MD trajectory (Appendix A and Results section).

For this reason, in the production runs, free energy calculations were based on configurations obtained from 10 replicas of a 2 ns equilibrium MD simulation under periodic boundary conditions at constant pressure. Configurations were extracted every 10 ps from the second half of the trajectory, accounting for a total of 1000 configurations and 10 ns of dynamics for each of the systems considered.

We computed the binding free energy (ΔG) for every configuration using MM-PBSA in the single-trajectory approximation [30] as ΔG = G_complex_ − (G_ligand_ + G_protein_). Every free energy term was calculated as an average over the considered structures: 〈G〉 = 〈G_MM_ 〉 + 〈G_Solv_〉, where the entropic term is omitted. The vacuum potential energy G_MM_ was calculated based on the molecular mechanics (MM) force-field parameters (G_Coul_ and G_VdW_). The solvation term G_Solv_ was split into polar and non-polar contributions (G_PolSolv_ and G_nonPolSolv_) and was computed with the Adaptive Poisson–Boltzmann Solver (APBS) program [44]).

The 10 different replicas could produce, in principle, significantly different trajectories, but we did not expect large deviations within each of the trajectories. For this reason, errors were calculated as standard errors of the set of the 10 different average ΔG coming from each of the simulations’ replicas, and the errors of ΔΔG were obtained from the formula of propagation of errors.

### 2.2. Monte Carlo Simulations

To force the sampling of sequences toward a binding energy minimum, we relied on the Monte Carlo multi-dimensional search method with the Metropolis algorithm [33]. Mutants were designed with the same protocol of the in vitro maturation performed in [4]: mutations were allowed only in the CDR3 of the heavy chain of the antibody excluding Gly100, Cys102, and Cys107. Furthermore, no amino acid could be mutated into Gly, Leu, Met, Trp, Tyr, Cys, or His. Given a starting sequence of the antibody (antibody S), we randomly selected with equal probability one of the residues and mutated it into another random residue, obtaining a new antibody (antibody M). We then computed the binding free energies between the antibody and the pepN9-19 using the MM-PBSA method based on the MD simulations described above. If the binding free energy of antibody M to pepN9-19 was lower than that of antibody S (ΔΔG = ΔG_M_ − ΔG_S_ < 0), M was accepted and became the new antibody S in the following Monte Carlo cycle. If the binding energy of antibody M was higher than that of S (ΔΔG > 0), antibody M was accepted with a probability equal to the Boltzmann factor of the variation of binding free energy (e^−ΔΔG/KT^). If the change was not accepted, antibody S kept its role in the next cycle.

### 2.3. Antibody Expression and Purification

DNA sequences of the corresponding VHs were cloned in the pFuse heavy-chain vector (#pfusess-hchg1, InvivoGen USA, San Diego, CA, USA) of a human IgG1 antibody. The VL sequence, which was the same for all studied antibodies, was cloned in the pFuse light-chain vector (#pfuse2ss-hcll2, InvivoGen USA, San Diego, CA, USA). Plasmids of the heavy-chain and light-chain vectors were co-transfected into 293F cells that were maintained in FreeStyle 293 medium (#12338-026, Thermo, Life Technologies Corporation, Grand Island, NY, USA) at 37 °C with 5% CO_2_, using 293fectin (#12347500, Gibco) according to the manufacturer’s instructions. The resulting cells were cultured for 5 days for antibody expression and secretion. The medium was harvested for purification with the HiTrap Protein A HP column (#17-0403-03; GE Healthcare Bio-Sciences AB, Uppsala, Sweden) using an ÄKTAxpress purifier (GE Healthcare Bio-Sciences AB, Uppsala, Sweden). Purified antibodies were concentrated and stored in PBS buffer (pH 7.4) at 4 °C for up to 1 month or at −80 °C for up to 1 year.

### 2.4. Measurement of Binding Affinity Using Surface-Plasmon-Resonance

Biotin-labeled human CXCR2 N-terminal extracellular peptide (pepN48, the antigen for all the antibodies, which corresponds to the first 48 residues in the CXCR2 protein) was immobilized on the streptavidin-coated surface of a Series S Sensor Chip SA (#BR100531; GE Healthcare, Cytiva, Uppsala, Sweden). A Biacore T200 system (GE Healthcare) was employed for the measurement. The antibodies were serially diluted to the indicated concentrations in a running buffer (HBS-EP + buffer pH 7.4 (BR100669; GE Healthcare)) and injected as the analytes. The kinetics of association/dissociation were measured and fitted to an appropriate protein–protein interaction model to calculate the corresponding binding constant (K_D_).

### 2.5. Measurement of the Antibody Binding to CXCR2 Expressed on Live Cell Surface

The Chinese hamster ovarian cell line (CHO-K1) was transduced with the lentivirus carrying a full-length human CXCR2 coding DNA sequence (NM_001557) under the EF1a promoter, after which a stable hCXCR2 expressing the CHO cell line was selected and used in this study. An empty CHO-K1 cell line was used as a control. The stable cell line and CHO-K1 were maintained in Ham’s F-12K medium (#21127022; Gibco Life Technologies Corporation, Grand Island, NY, USA) containing 10% (*v*/*v*) FBS at 37 °C with 5% CO_2_.

Cells cultured at about 70% confluency were detached using 0.01% trypsin and washed with PBS. Then, 5 × 10^5^ cells were incubated with 2 μg/mL of each antibody in 100 μL of PBS with 0.5% BSA for 15 min on ice followed by a PBS wash. Secondary Alexa Fluor 488 antibody binding to human IgG (#A11013, Invitrogen, Life Technologies Corporation, Eugene, OR, USA) was used for flowcytometry detection (1:1000) according to the manufacturer’s instructions (CytoFLEX S, Beckman Coulter Life Sciences, Suzhou, China).

## 3. Results

### 3.1. Computed Binding Free Energies Correlated with Experimental Binding Affinity

To provide a proof of concept that in silico affinity maturation is possible in a practical setup, we considered the pool of antibodies obtained in [4] by phage panning and in vitro affinity maturation using a peptide corresponding to the first 48 residues of the N-terminus of CXCR2 (pepN48) as bait.

Calculating free energies is a mature tool that can produce reliable results, but it also has well-known shortcomings. Its sensitivity might not be high enough to distinguish between data points with low binding-affinity differences. The quality of the results is possibly system-dependent and strongly depends on the quality of the initial model, especially if experimental structures are not available.

It is commonly accepted that an accurate sampling of the configuration space is necessary to produce reliable results. This was particularly true in our case because mutated residues needed to find a new equilibrium position.

The sampling is generally done using MD simulations; however, the accuracy of the results depends on their length, as MD trajectories can become trapped in local minima that do not correspond to the most probable configuration. 

Our method of choice for free energy calculations was MM-PBSA [44], which provided reliable results in our previous study on an antibody–antigen system [38].

Previous works reported that when using the MM-PBSA method for calculating the binding free energy, longer simulations tended to produce less accurate results [32,45,46], but this is not a general rule, as it can depend on the system in consideration [47].

For this reason, we tested the accuracy of MM-PBSA calculation using two different sampling schemes. In the first case, we extracted 2000 configurations from the last 20 ns of a single 25 ns MD trajectory; in the second case, we produced 10 replicas of a 2 ns trajectory and extracted 100 configurations from the last 1 ns of each of them for a total of 1000 configurations. Comparison with the experimental binding affinity (Appendix A) showed that both sampling schemes produced results with a fair correlation with experimental data, but the latter was slightly better. Therefore, for the data production runs, we abandoned longer simulations in favor of the second sampling scheme.

Despite the fact that all the experimental binding affinities were on the order of nano- or sub-nanomolar, we noticed that the computed binding energy was precise enough to distinguish between high-affinity and very-high-affinity antibodies (Figure 1 and Table 1) and that the correlation was sufficiently high to use computer simulations as a screening methodology (R^2^ = 0.57, Table 2). It is also important to note that in the simulation setup, we considered only a shorter version of the bait peptide (pepN9-19) that was used to obtain the crystal structure of the antibody-peptide complex and that we considered only the FAB portion of the antibody, while in the experiments, the antibodies were in the full-length IgG1 format.

The primary computational bottleneck in the proposed procedure was calculating the binding free energy using the MM-PBSA algorithm, which in our MD dedicated GPU cluster took 8 h for each GPU node. The method scales linearly with the number of nodes but would still not be feasible for researchers with limited computational resources. We recently tested the accuracy of the PRODIGY web server [38,45,46] in predicting binding affinities and found that we could qualitatively reproduce the results but with a much-reduced computational cost. However, the quality of the results appeared to be system-dependent because the binding score of the PRODIGY web server for the antibody against CXCR2 did not correlate with the experimental data (R^2^ = 0.07) and poorly correlated with the binding energy computed using the MM-PBSA method (R^2^ = 0.32, Figure 2).

In an attempt to understand the differences at the molecular level between the various antibodies, we present a detailed analysis of abN48 and abN48-2 antibodies in Figure 3. The latter was chosen as the more promising candidate for follow-up experimental work due to its biochemical properties, especially when converted into the IgG format [4]. The MM-PBSA calculations showed that the mutations of the two serines in arginines improved the binding affinity, as expected from the experiments.

However, it was difficult to pinpoint the contribution of these mutations by looking at the structure alone or even after a careful analysis of the trajectories, as the two arginines are not directly interacting with the peptide. The more significant difference between the two antibodies appears to be the increased flexibility of the antibody CDR regions for abN48-2, as shown by the root-mean-square fluctuation (RMSF) spectra in Figure 3C,D. As a consequence, the peptide can fit better into the binding pocket and improve its electrostatic interaction with the antibody (−3809 kJ/mol for abN48-2 vs. −2971 kJ/mol for abN48). On the other hand, the non-polar solvation contribution to the binding energy become worse (+3760 kJ/mol for abN48-2 vs. +3015 kJ/mol for abN48). Overall, these energy contributions account for a net improvement of ΔΔG = −123 kJ/mol for the maturated antibody. Such differences are very difficult to predict, even by experienced structural or computational biologists, and make the design of a similar antibody virtually impossible using only human intuition.

### 3.2. Monte Carlo Sampling of the Sequence Space Rapidly Converged to Antibodies with Improved Affinity

In addition to producing accurate binding energy predictions, it is equally important to explore the incredibly large space sequence to ensure that improved antibodies can be found in a reasonable amount of time. While the procedure can be easily parallelized and scaled up by increasing the number of nodes in a computational cluster, MM-PBSA remains a computationally expensive tool. As a consequence, only a few antibody sequences can be tested with the desired accuracy per day per single node.

To explore the chemical space in a sensible way and reduce the risk of falling into entropic traps, we decided to generate Markov chains [48] of antibody sequences using a Metropolis algorithm aimed at maximize the affinity between the antibody and the target peptide. As we explained in the Materials and Methods section, at each step of the Monte Carlo simulation, a new sequence was derived from the previous by introducing a single point mutation on the CDR3 of the antibody. The new antibody’s binding energy was computed using the MM-PBSA algorithm; the new antibody replaced the previous one if it was a better binder or with a probability proportional to the Boltzmann factor of the energy difference if it was worse (Figure 4A).

We ran three replicas of the Markov chain simulation, each starting from the same initial antibody (abN48). Each replica lasted for 26 attempted steps (corresponding to about 10 days of simulations on three different nodes of the computational cluster, a time comparable with that necessary to perform antibody maturation in vitro). Due to the stochastic nature of the Monte Carlo method, the replicas produced sequences significantly different from the initial one after the very first step of the Markov chain. In all three cases, the Markov chain rapidly found an antibody sequence with an improved binding energy, despite the starting point already being in the sub-nanomolar affinity (Figure 4B). In most cases, a single point mutation produced an antibody with a binding affinity close to the previous one in the Markov chain. Still, it was also possible to find particularly favorable or unfavorable mutations. The values of ΔΔG of the attempted mutations are skewed toward negative values (Figure 4D), a result expected from a Metropolis algorithm that had not yet converged to a minimum. The antibodies generated in this way appears to belong to an entirely new branch in the phylogenetic tree if compared with those produced using in vitro maturation (Figure 4C), indicating that in silico maturation could be used in parallel to the experimental methods to improve the sampling of the sequence space.

### 3.3. In Vitro Test of Antibodies Showed the Efficacy of the Method

Finally, we wanted to test whether the newly designed sequences were better binders also in the experiments. This was done using the Surface Plasmon Resonance (SPR) method to test the binding between the antibodies and the N-terminal peptide. The results showed that two out of three antibodies (S1-M22 and S2-M12) were better binders compared to abN48, both with a binding affinity (K_D_ = 4.6 × 10^−10^) about three times better than that of the abN48 antibody (K_D_ = 1.3 × 10^−9^) and in the same order of magnitude of the best antibody obtained from the experiments (abN48-2, K_D_ = 1.1 × 10^−10^, Figure 5A and Table 3). More importantly, flowcytometry experiments showed that both antibodies could recognize and bind to CXCR2 proteins expressed on the Chinese hamster ovarian (CHO-K1) cell surface (Figure 5B and Table 3). Instead, the third in-silico-designed antibody did not bind to the pepN48 peptide in the SPR experiment, but displayed some residual binding on the CHO-K1 cells that expressed CXCR2. This could indicate that the third antibody had poor biophysical properties or a binding affinity much lower than predicted from the simulations. Even though the affinities of the two functioning antibodies to the peptide were the same, the K_ON_ and K_OFF_ were notably different, showing that the three Markov chains found three antibodies that were very different in their biochemical properties.

## 4. Discussion

In this article, we tested the possibility of substituting an antibody-maturation experiment with computer simulations under the requirement of using a similar timeframe to produce the results. There is no lack of applications for such methodology, as computer simulations are considerably cheaper than experiments and can also be used for targets that are difficult to study experimentally, leading to an increasing payoff for the pharmaceutical industry [49].

As our test model, we chose an antibody characterized in a previous work by our collaborators [4] that binds to the CXCR2. This system was chosen because its structure is known and because we already had preliminary data to test the quality of our predictions. It remained a challenging task, as the starting antibody already has a good affinity (in the nanomolar range) to its peptide target.

We faced two critical problems, and both had to be solved to produce reliable results, i.e., the ability to accurately compute the binding free energy in a reasonably short amount of time and the ability to explore the space of antibody sequence in a sensitive way.

We showed that it is possible to obtain a good correlation between experimental binding affinities and theoretical values for the ΔΔG even within the small range of the explored K_D_. 

More importantly, we managed to refine the starting sequence in about 20 random attempts using a Monte Carlo method based on the Metropolis algorithm. While one of the predicted antibodies did not appear to bind to the target at all, the other two improved the binding affinity threefold and were comparable to the best one selected by the in vitro methodology. The sequences produced by our computational methodology are very different from the experimental ones in the phylogenetic tree. This could indicate that good binders to a given target are not rare and that we can use computational approaches to complement experimental methodologies, increasing the chances that one of the hits has the desired biochemical and biophysical properties. 

Our methodology can be improved and refined in future iterations. MM-PBSA is not the only possible choice for the computation of the binding free energies. Other methodologies such as alchemical transformation, free energy perturbation (FEP) [50,51] or computing the potential of mean force (PMF) necessary to pull apart the two proteins [28,51,52] have proved their feasibility, and can be used as valid alternative.

It is also possible to use simplified methods based on coarse-grained force fields [52,53,54] to obtain a faster evaluation of the binding free energy or, vice versa use more computationally expensive quantum chemistry calculations to obtain more accurate predictions [55,56]. In a recent work [38], we showed that it was possible to find a good correlation between the results obtained using the PRODIGY web server [57] and MM-PBSA results [44], but this was not the case for the system of this study, indicating that the results of simplified methods could be system-dependent. The difference in the performance could be due to the lack of sensitivity of the PRODIGY score function in the range of the experimental K_D_ or because the interaction between the antibody and the peptide was mainly due to hydrophobic contacts. Unfortunately, this also means that the accuracy of similar approximated methodologies needs to be experimentally validated for each specific system, de facto eliminating the advantage of a purely computational method.

Despite the correlation between the simulations and experiments not being perfect, the methodology we proposed was revealed to be precise enough to clearly distinguish antibodies with a very high affinity from those with a lower affinity. Screening methods are not meant to be extremely precise. False positives and negatives produced by a screening method are well tolerated as long as the same method can also produce true positives that can move on in the experimental pipeline. Indeed, both experimental and computational determinations of binding affinities are prone to high relative uncertainties. Computed or measured binding affinities only indicate the possibility that an antibody will bind to the target on the cell surface. These techniques are meant to increase the number of hits before obtaining the best lead for the follow-up experiments. It is also important to note that an excellent binding affinity is only the first step toward selecting a valuable antibody for therapeutic purposes; the more suitable antibody should have other important biochemical properties (e.g., stability, sensitivity, or specificity to the desired target, and so on) before becoming a biotechnological tool or a real drug candidate.

## Figures and Tables

**Figure 1 biomolecules-12-01285-f001:**
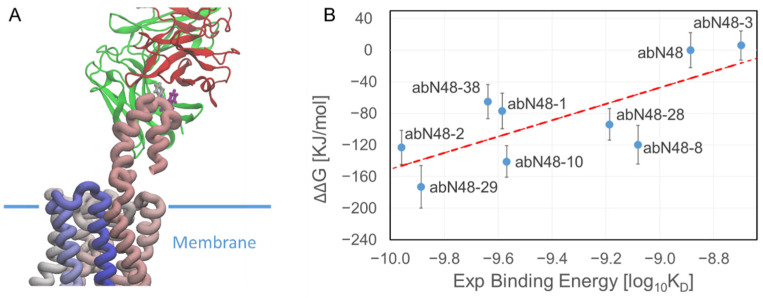
Correlation between experimental binding affinity and computed binding free energy. (**A**) Illustration of the abN48-CXCR2 system. The light chain and heavy chain of abN48 in FAB format are shown in in red and green, respectively. The CXCR2 protein is shown as a ribbon; the color goes from pink (n-terminal) to blue (c-terminal). Phe14 (gray) and Trp15 (purple) of CXCR2 are the main contributors to the interaction with the pepN9-19. The structure of the N-terminal of CXCR2 is not known except for the crystalized residues and is modeled here as a random coil. (**B**) Correlation between the binding affinity ΔΔG and the log_10_ of the experimental K_D_. The correlation coefficient between the two sets of data was R^2^ = 0.57.

**Figure 2 biomolecules-12-01285-f002:**
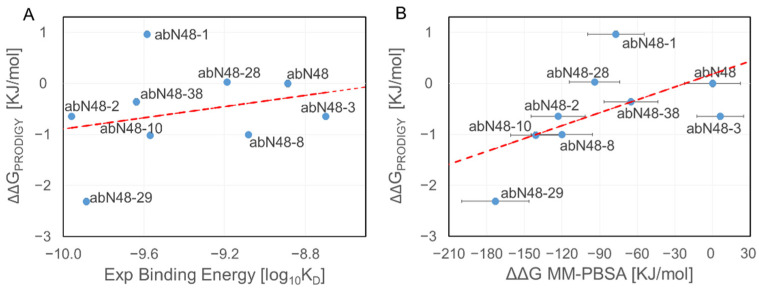
Binding affinity using the PRODIGY web server. The binding affinity of abN48 and all its mutants in complex with pepN9-19 was computed using PRODIGY. Calculations were done using the same configurations used in the MM-PBSA method. Results obtained with PRODIGY web server did not correlate with the log_10_ of the experimental K_D_ ((**A**), R^2^ = 0.07) and poorly correlated with MM-PBSA results ((**B**), R^2^ = 0.32).

**Figure 3 biomolecules-12-01285-f003:**
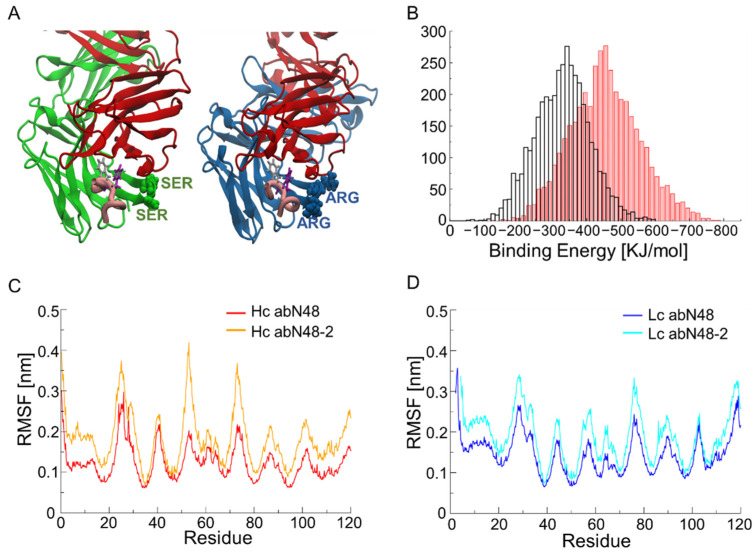
Details of molecular dynamics simulation of abN48 and abN48-2. (**A**) Illustration of abN48 and abN48-2 FABs and their interaction with pepN9-19 (colored in pink; Phe14 and Trp15 are shown in gray and purple, respectively). The two antibodies shared the same light chain (in red) while the heavy chains differed only for two residues (two serines of CDR3 of abN48 were mutated in arginines in abN48-2). (**B**) Histogram of the binding energy computed using MM-PBSA on the abN48-pCXCR2 complex (black) and abN48-2-pCXCR2 complex (red). The two distributions had clear separate picks and the *t*-test showed a *p*-value < 0.05. Root-mean-square fluctuation (RMSF) of abN48 and abN48-2 heavy chains (**C**) and light chains (**D**). The maturated antibody showed higher fluctuations than the original one, in agreement with the β factor of the crystal structure.

**Figure 4 biomolecules-12-01285-f004:**
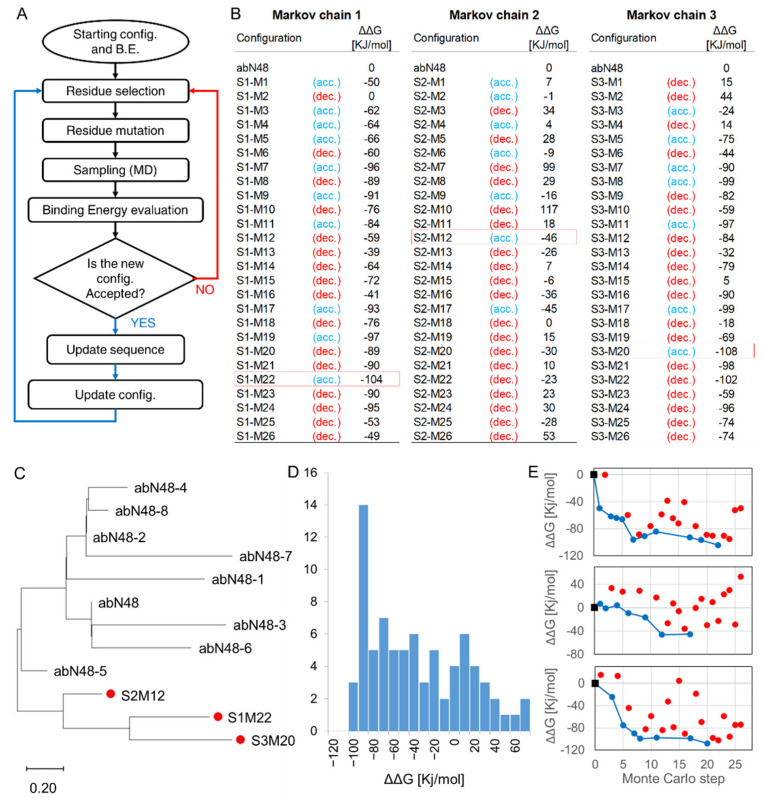
In silico affinity maturation predictions. (**A**) Schematic representation of the Metropolis algorithm. At each step, a new sequence was proposed; if the binding energy of the new configuration was lower than that of the previous one, the new configuration was always accepted; otherwise, the new configuration was accepted with a probability equal to e^−ΔΔG/KT^. (**B**) data obtained using the three Markov chain replicas: at each step we indicated whether the mutation was accepted (acc.) or declined (dec.). The best binders for each Markov chain are highlighted with red borders and were selected for experimental validation. (**C**) Maximum likelihood phylogenetic tree constructed using MEGA version 11. The scale bar below indicates the distance (amino acid substitution per site). In silico designed antibodies are represented with red dots. (**D**) Histogram of all the ΔΔGs computed during the three Markov chain repetitions. (**E**) Time course of the energy evaluated during the three different Markov chain repetitions. The starting points are represented with black squares; the energy is shown in blue if the mutation was accepted and in red if it was declined.

**Figure 5 biomolecules-12-01285-f005:**
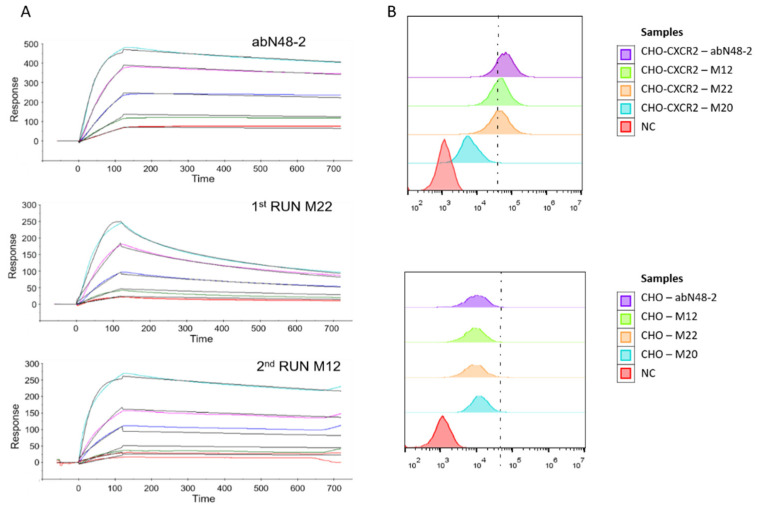
Experimental validation of theoretical predictions. (**A**) Binding kinetics and affinity of abN48-2, S1-M22, and S2-M12 to the peptide of extracellular N-terminal domain of human CXCR2 were determined using the SPR method (Biacore T200 system). Lines of different colors correspond to raw data obtained for different antibody concentrations (red curve: 0.5 nM, green curve: 1 nM, blue curve: 2 nM, magenta curve: 5 nM, cyan curve: 10 nM), while the black lines correspond to the best global fit). S3-M20 binding to the antigen was not detected in the experiment. (**B**) The binding ability of abN48-2, S1-M22, S2-M12, and S3-M20 to the native human CXCR2 on the surface of CXCR2-overexpressing CHO cells were evaluated using cytoflowmetry (CytoFlex). The bottom panel represents the control experiment with CHO cells not expressing CXCR2. Histograms represent the distribution of cells bound with antibodies as cell number vs. the amount of bound antibody (FL intensity—fluorophore-labeled antibody).

**Table 1 biomolecules-12-01285-t001:** Different contributions to the binding for the various antibodies produced by in vitro affinity maturation. Absolute values for ΔG were obtained from the sum of the other contributions; the relative ΔΔGs were obtained as differences with the absolute ΔG of abN48.

	ΔΔG_MM-PBSA_	ΔG_MM-PBSA_	ΔG_coul_	ΔG_vdw_	ΔG_PolSolv_	ΔG_nonPolSolv_
	(kJ/mol)	(kJ/mol)	(kJ/mol)	(kJ/mol)	(kJ/mol)	(kJ/mol)
abN48	0 ± 22	−323 ± 16	−2971 ± 42	−334 ± 5	3015 ± 35	−33 ± 0.5
abN48-1	−77 ± 23	−400 ± 16	−3849 ± 57	−360 ± 3	3845 ± 47	−36 ± 0.4
abN48-2	−123 ± 21	−446 ± 15	−3809 ± 48	−362 ± 2	3760 ± 38	−36 ± 0.3
abN48-3	6 ± 14	−317 ± 10	−3034 ± 36	−347 ± 4	3099 ± 36	−35 ± 0.3
abN48-8	−120 ± 26	−443 ± 19	−3851 ± 45	−354 ± 3	3799 ± 39	−36 ± 0.4
abN48-10	−141 ±17	−464 ± 12	−3861 ± 46	−356 ± 3	3790 ± 42	−36 ± 0.4
abN48-28	−94 ± 18	−417 ± 12	−3499 ± 47	−367 ± 3	3484 ± 42	−35 ± 0.4
abN48-29	−173 ± 31	−496 ± 22	−4196 ± 66	−348 ± 2	4083 ± 50	−35 ± 0.3
abN48-38	−65 ± 21	−388 ± 15	−3751 ± 54	−351 ± 2	3749 ± 51	−36 ± 0.4

**Table 2 biomolecules-12-01285-t002:** Comparison between the ΔΔGs evaluated with the MM-PBSA method or PRODIGY web server and the experimental values of the K_D_.

	Exp. Values	ΔΔG_MM-PBSA_	ΔΔG_PRODIGY_
	(M)	(kJ/mol)	(kJ/mol)
abN48	1.3 × 10^−9^	0 ± 22	0.00 ± 0.07
abN48-1	2.6 × 10^−10^	−77 ± 23	0.97 ± 0.06
abN48-2	1.1 × 10^−10^	−123 ± 22	−0.64 ± 0.06
abN48-3	2.0 × 10^−9^	6 ± 19	−0.65 ± 0.06
abN48-8	8.3 × 10^−10^	−120 ± 24	−1.00 ± 0.06
abN48-10	2.7 × 10^−10^	−141 ± 20	−1.01 ± 0.06
abN48-28	6.5 × 10^−10^	−94 ± 20	0.03 ± 0.06
abN48-29	1.3 × 10^−10^	−173 ± 27	−2.31 ± 0.06
abN48-38	2.3 × 10^−10^	−65 ± 22	−0.36 ± 0.06

**Table 3 biomolecules-12-01285-t003:** Experimental binding assays on the antibodies resulting from in silico maturation. K_ON_, K_OFF_, and K_D_ values were obtained using SPR. After fitting the data in Figure 1, the predicted values for the K_D_ were 2.8 × 10^−10^, 1.0 × 10^9^, and 2.6 × 10^−10^ for S1-M22, S2-M12, and S3-M20, respectively. The geometric mean fluorescence intensity and positive cell percentage (gated by the dashed line) of each group shown in Figure 5B are shown in the last two columns.

	K_ON_ (1/Ms)	K_OFF_ (1/s)	K_D_ (M)	Mean Fluorescence Intensity	% of Positive Cells
abN48-2	2.5 × 10^6^	2.7 × 10^−4^	1.1 × 10^−10^	174.7 × 10^3^	98.6
S1-M22	4.0 × 10^7^	1.8 × 10^−2^	4.6 × 10^−10^	40.8 × 10^3^	40.9
S2-M12	7.2 × 10^5^	3.3 × 10^−4^	4.6 × 10^−10^	43.1 × 10^3^	43.0
S3-M20	n/a	n/a	n/a	6 × 10^3^	0.2

## Data Availability

The data presented in this study are available upon request from the corresponding author. The two newly discovered antibody sequences presented in this paper cannot be disclosed due to legal agreements.

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
