# Peer review of "In Silico Maturation of a Nanomolar Antibody against the Human CXCR2"

_biomolecules, 2022, doi:10.3390/biom12091285_

Round 1
Reviewer 1 Report
The manuscript biomolecules-1844565, titled “In silico maturation of a nanomolar antibody against the human CXCR2”, proposes and evaluates a computational method for the design of novel, more potent antibodies against a specified target, by introducing mutations. The authors propose a protocol involving short molecular dynamics simulations and energy calculations to evaluate the binding affinity of the mutated antibodies, as well as a Metropolis/Monte Carlo - based approach to predicting mutation capable of increasing antibody affinity. The proposed methods are tested using the abN48 antibody - CXCR2 N-terminus complex as a case study and are coupled with experimental assays, that show that two out of the three proposed mutated antibodies do have an increased affinity for their target.
Overall, this is an interesting work, presenting a very intriguing protocol, especially regarding the Monte Carlo design pipeline. What is more, the paper is written in very good English, and the methodology is presented in adequate detail. However, while experimental validation showcases, to some extent, the potential of the proposed method, I have some concerns, especially regarding the MD simulations and energy calculations parts.
Major:
Point 1. My main concern lies with the authors using very short simulations to analyze their data. While the equilibration part of their protocol is solid, they only perform 1 nanosecond of production simulation. In such short production MD timescales, the proteins in the system are expected to be unconverged and actively evolving towards their final, more stable conformation. For medium sized protein systems of good quality, this is typically around 25-30 ns, while for larger, more complex systems, it may require a lot more. This is expected, both due to the artifacts of the original structure, and due to the nature of simulations themselves. The former may be mitigated by using a high quality (i.e. <2.0 angstroms resolution, minimum to no disorder) structure; however these are rather rare. In fact, most antibody structures in the PDB are in the 2.5-3.5 angstroms range.
If the authors have the available computational resources, I would advise them to increase their simulation times to at least 25 ns per system and use the final 1 ns of each trajectory in the analysis. Alternatively, they could perform a long simulation of the original WT complex to reach convergence, introduce mutations to the converged system, and then perform and analyze short simulations of the mutants from the converged structure.
If the computational resource to do the above are no longer available, the authors should provide a segment discussing this limitation. In either case, the authors should provide evidence (e.g. RMSD-based measurements) of the systems’ progression during MD.
Point 2. The authors have used MM-PBSA to perform binding energy calculations. The results are given in Tables 1 and 2, both accumulated, and decomposed to each energy component. However, only absolute values are given, presumably the mean averages of each calculation. Considering the fact that these come from very short simulations (as I already discussed in Point 1), AND the fact that MM-PBSA has the tendency to produce high error values, especially for the Coulomb and Poisson-Boltzmann parts, the authors are advised to also provide the confidence intervals (standard deviation or standard error) of their calculations, in the form of average +/- mean/error.
Point 3. As the authors themselves state, MM-PBSA calculations are computationally taxing, especially for the PBSA part (in their case, performed with APBS). Have they considered using other free energy calculation methods, such as Free Energy Perturbation or Thermodynamic Integration, or Potential of Mean Force (PMF) methods using steered MD, such as Umbrella Sampling or the Jarzynski Equality? A lot of these methods produce results a lot closer to experimental measurements, compared to MM-PBSA, and they also take into account the overall dynamics of the system (which means that entropy, the PV contribution etc. can also estimated), leading to even more accurate predictions. In addition, they can be a lot faster than Poisson-Boltzmann calculations, especially when coupled with advanced sampling techniques (e.g. replica exchange) or even simple tricks to enhance sampling (e.g. slightly increasing the temperature to provide more kinetic energy). Perhaps the authors could add a short segment discussing alternatives to MM-PBSA and, in future works, compare the various approaches both in terms of accuracy, and in terms of computational efficiency.
Point 4. I found the Monte Carlo approach for identifying mutations and designing antibody variants especially interesting. Are there plans to implement this as a tool or pipeline?
Minor Point: In the first paragraph of the introduction, the orders of magnitude given seem to be mis-typed or mis-formatted (lines 34 and 37, read “...on the order of 1020” and “...the order of 5 × 109”). The authors should find and correct this and any other similar typos.
Reviewer 2 Report
The goal of the article "In silico maturation of a nanomolar antibody against the human CXCR2" is to implement an in silico approach to assess the impact of variants during the maturation of antibodies using structural information. The in silico methodology mentioned in this manuscript has been explored in the past with variations on the sampling, mutation and scoring phases. Something important is to highlight these previous works and its impact with antibodies, peptides and protein fragments (https://link.springer.com/protocol/10.1007/978-1-0716-1855-4_16). Based on this, there are multiple factors that are not clear in the manuscript and rise major concerns that need to be addressed: Major points: 1. As mentioned in parts of the manuscript, the MD simulation is crucial. For the assessment, and based on the Methods, only 1 ns was simulated. This is a very short time for a protein-protein complex and should be increased and supported with RMSD metrics of the complexes stability. In addition, the same 1 ns was used during the Monte Carlo design process? This is not mentioned and still it is a very short sampling time that should be corrected. 2. The benchmark suffers from different issues. First the affinity differences are very small between the variations based on Table 2. This is a big problem to assess given the intrinsic errors of techniques like MM-PBSA. Second, the standard deviations of the calculations are not reported, and based on Table 2 this will definitely impact the calculated correlation. Third, why are the experimental values on Table 1 different from Table 2, are those different variants? why the same code?. Fourth, the variants are not explicitly mentioned, and this is also a big deal given how the type of mutation can affect the binding affinity. 3. The PRODIGY calculations are not improving the analysis. The correlation shown in Figure 2 shows that this is not providing any connection with the experimental values, and should be discarded for any design decision made in the manuscript 4. The experimental part is disconnected from the analysis. Only two variants were shown and it is not clear what are the calculated KD values and what was the correlation with the in silico simulations (given the experiments shown in SI). Were the final accepted peptides subjected to longer MD simulations and recalculated MM-PBSA values? The Figure 5 is confusing and not statistical differences are provided to decide which antibodies are better Minor concerns: 1. Format issues with the author names, Table legends and Figure resolutions 2. The SI Figure requires a captionAuthor Response
Please see the attachment.

Reviewer 3 Report
Summary
Buratto et al. used molecular simulation, namely Monte Carlo Metropolis coupled with Molecular Mechanics/Poisson-Boltzman surface area, to design a nanomolar antagonist against a Chemokine Receptor 2. They showed that the affinity of the newly designed sequences are comparable to the affinity of a benchmark in vitro designed sequence. The work is of interest to the field of antibody engineering and serves as a proof principle for the in silico design of tightly binding antibodies
Comments
-
Many formatting errors throughout the manuscript, e.g., line 36 1016–1018 etc
-
In line 247, the authors mentioned increased flexibility of the CDR regions. How did the authors annotate these regions, was an antibody numbering scheme used? If so, which scheme? See for example https://doi.org/10.1016/j.celrep.2021.108856 where structures were obtained from https://doi.org/10.1093/database/bay040 and numbered with the martin scheme.
-
Mixed up Figure 4C and 4D, line 286
-
Avoid using red and green together in the plot (suboptimal for color blind readers)
-
In line 315, the authors mentioned that the third in silico-designed antibody did not bind to the original epitope. We ask the reviewer to elaborate more on this and to contrast and discuss the impact of epitope discrepancy on phenotypic behaviour
Round 2
Reviewer 1 Report
The authors have addressed my concerns, as well as the concerns of the other reviewers. The manuscript, after these revisions, can be ACCEPTED for publication.
Author Response
We thank Reviewer 1 for helping us improving our work.